

# Children's attitudes towards animals are similar across suburban, exurban, and rural areas

Stephanie G. Schuttler[1], Kathryn Stevenson[2], Roland Kays[1,3] and Robert R. Dunn[4,5]

[1] North Carolina Museum of Natural Sciences, Raleigh, NC, USA
[2] Department of Parks, Recreation and Tourism Management, North Carolina State University, Raleigh, NC, USA
[3] Department of Forestry & Environmental Resources, North Carolina State University, Raleigh, NC, USA
[4] Department of Applied Ecology, North Carolina State University, Raleigh, NC, USA
[5] Center for Macroecology, Evolution and Climate, Natural History Museum of Denmark, University of Copenhagen, Copenhagen, Denmark

Corresponding author
Stephanie G. Schuttler,
stephanie.schuttler@naturalsciences.org

## ABSTRACT

The decline in the number of hours Americans spend outdoors, exacerbated by urbanization, has affected people's familiarity with local wildlife. This is concerning to conservationists, as people tend to care about and invest in what they know. Children represent the future supporters of conservation, such that their knowledge about and feelings toward wildlife have the potential to influence conservation for many years to come. Yet, little research has been conducted on children's attitudes toward wildlife, particularly across zones of urbanization. We surveyed 2,759 4–8th grade children across 22 suburban, exurban, and rural schools in North Carolina to determine their attitudes toward local, domestic, and exotic animals. We predicted that children who live in rural or exurban areas, where they may have more direct access to more wildlife species, would list more local animals as "liked" and fewer as "scary" compared to children in suburban areas. However, children, regardless of where they lived, provided mostly non-native mammals for open-ended responses, and were more likely to list local animals as scary than as liked. We found urbanization to have little effect on the number of local animals children listed, and the rankings of "liked" animals were correlated across zones of urbanization. Promising for conservation was that half of the top "liked" animals included species or taxonomic groups containing threatened or endangered species. Despite different levels of urbanization, children had either an unfamiliarity with and/or low preference for local animals, suggesting that a disconnect between children and local biodiversity is already well-established, even in more rural areas where many wildlife species can be found.

## INTRODUCTION

One of the biggest threats to the conservation of biodiversity is the "extinction of experience," a term used to describe the largescale decline of people's time spent in nature

and the diverse experiences time in nature entails (*Pyle, 1978*). Individuals who have had more experiences are more likely to have pro-environmental attitudes, especially when those experiences occurred during childhood (*Soga & Gaston, 2016*). Today, children spend much less time outdoors than the generations before them, and fewer people live in rural areas surrounded by large, natural spaces (*Kellert et al., 2017*). As people spend more time indoors and have less access to natural areas in their daily lives, their familiarity with and perspectives toward local wildlife will likely change.

While many conservation biologists focus on challenges associated with the health of an ecosystem such as habitat loss, declines in native biodiversity, increases in invasive species, and pollution (*Aronson et al., 2014*; *Dirzo et al., 2014*), challenges that relate to societal perceptions are equally important. These perceptions of nature set the template that influences the future willingness of the public to invest in the conservation of nature. For example, the intolerance of wildlife, perceived threats or nuisances, and a lack of funding and public support for policy can all thwart otherwise-successful conservation efforts (*Brook, Zint & De Young, 2003*; *Inskip & Zimmermann, 2009*). It is especially pertinent to study the perceptions children have on wildlife, as they are the future stakeholders, and interventions made during childhood are more likely to be successful when values are still forming (*Feinsinger, 1987*; *Manfredo et al., 2017*).

Children tend to favor what have been termed "loveable animals," which includes domestic pets and large, charismatic megafauna (*Bjerke, Ødegårdstuen & Kaltenborn, 1998*; *Borgi & Cirulli, 2015*; *Lindemann-Matthies, 2005*). In fact, some pets can have a negative impact on wildlife (*Doherty et al., 2017*; *Loss, Will & Marra, 2013*). Many species of charismatic megafauna are of conservation concern, such as the case for pandas, great apes, big cats, elephants, and rhinoceros. Indeed, to the extent that megafauna are often not only threatened, but also conservation targets (*Dietz, Dietz & Nagagata, 1994*; *Smith & Sutton, 2008*), the fondness of children for "loveable animals" may actually lead to a fondness for species of conservation concern. One recent study even showed potential for children to align more closely with conservationists' prioritization of species attributes (*Frew, Peterson & Stevenson, 2016*) than adults in a similarly designed study (*Meuser, Harshaw & Mooers, 2009*).

Yet, if children only value charismatic megafauna and pets, they may lose connections to local species and hence a willingness to conserve species nearby. One arbiter of whether children value local species may be their experience with those species (*Ballouard, Brischoux & Bonnet, 2011*; *Lindemann-Matthies, 2005*; *Schlegel & Rupf, 2010*). Given that more children live in urban or urbanized landscapes than in previous generations, and that urbanization impacts the richness and diversity of wildlife communities (*McKinney, 2008*), a "pigeon paradox" (*Dunn et al., 2006*) may occur, where people will be motivated to protect species they are most familiar with, but in places where those species tend to be common and pest species. Under this scenario, people will primarily experience nature through these common and even invasive, urban species (which only rarely need conservation attention) or virtually through the Internet and television (where the focus is often on exotic megafauna). Whether this is the case in practice is unclear. Children in some regions have been shown to prefer species they never experience in real life

(*Ballouard, Brischoux & Bonnet, 2011*) and struggle to identify local wildlife compared to exotic species popularized in the media and even imaginary Pokémon characters (*Ballouard, Brischoux & Bonnet, 2011*; *Balmford et al., 2002*; *Genovart et al., 2013*). Whether the preferences of children for particular species varies with the degree of urbanization of their home place is unknown.

Additionally, modern lifestyle can even play a role in children viewing the outdoors negatively. One study found that children who had a stronger desire for modern comforts and manicured parks, had a dislike of wild, more natural spaces (*Bixler & Floyd, 1997*), whereas another found that some children in the UK even viewed wooded areas as "scary places" (*Milligan & Bingley, 2007*). With an increase in modern lifestyle, children today may be viewing nature as more scary, which could carryover to wild animals as well. For example, urban children in Norway viewed wolves and eagles as significantly more scary and dangerous than rural children did (*Bjerke, Ødegårdstuen & Kaltenborn, 1998*).

We investigated the preferences of 9–14 year old children toward wildlife, specifically animals, across different levels of urbanization in North Carolina, USA. We were interested in which animals children considered to be positive (i.e., liked), which were viewed negatively (i.e., scary) and whether these rankings were associated with how likely the children were to be able to experience these animals in their everyday lives (as a function of whether they were domestic, local, or exotic animals). Our objectives were to (1) identify the animals children recalled, (2) determine if children listed different animals for those they liked and those they considered scary, (3) categorize liked or scary animals as local, domestic, or exotic, and (4) understand how these categorizations varied across the level of urbanization (suburban, exurban, or rural) of the children' school and other demographic and socioeconomic factors. Due to presumed increased opportunities for encounters with animals in rural and exurban areas (*Zhang, Goodale & Chen, 2014*), we predicted that children from these schools would include more local species for liked animals and fewer for scary than children from suburban schools. In zones of higher urbanization (i.e., suburban areas), we predicted that children would favor non-native animals, as we expected their relationship with wildlife to be primarily based on virtual encounters, zoos, or pets.

## MATERIALS AND METHODS

### Sampling plan

We surveyed 4th, 6th, 7th, and 8th grade children in classrooms of teachers participating in the eMammal citizen science camera trap program (eMammal.org) in North Carolina from 2014 to 2017 (*Schuttler et al., 2017*, *2018*). North Carolina teachers were recruited through program advertisements, direct emailing, word of mouth, and through presentations at conferences. As this research was part of a study on the potential impacts of eMammal citizen science in the classroom, we also invited teachers from different schools within the same school districts to participate in the surveys, even if they would not be participating in the eMammal program. Surveys for this study were conducted prior to any mention of or implementation of the eMammal program. We asked participating teachers to include their children in the study by administering surveys in their classrooms.

Although teachers were self-selected into this study, which may relate to their perceptions of wildlife, children were included in the study based on their assignments to teachers, which relied on factors not related to wildlife perceptions or experiences in nature.

## Survey design and data collection

The questionnaire asked children to free-write four animals they liked most, four they found most scary, and to rank their top five favorite mammals from a list of 20. Children were asked the free list question first, but could see the rank question. As children do not think of animals based on biologists' taxonomic classifications, we named species or groups of species at the taxonomic level they would be able to identify (*Ballouard, Brischoux & Bonnet, 2011*). For the list of 20 mammals, we tried to pair local animals with exotic animals that children would know. While this included a list of more charismatic, exotic species compared to local ones, we did not expect local species to outrank charismatic, exotic species. Rather, we were interested to see where local species ranked amongst those that are well-known and liked by children and to determine if there are differences in rankings between children from different levels of urbanization. The list included 11 mammals local to North Carolina and nine exotic mammals (local: bobcat, coyote, raccoon, skunk, deer, rabbit, opossum, fox, bear, squirrel, bat; exotic: kangaroo, zebra, lion, panda, rhinoceros, monkey, wolf, whale, hedgehog). We defined local species in the context of children' ability to see a species in their daily lives and therefore included non-domestic, non-marine species (all schools were inland) with current range in North Carolina. Further, while red wolves do exist in North Carolina, they are restricted to a small range far from the schools surveyed in this study, so we categorized wolves as exotic. Survey questions are in Fig. S1.

During year one, we asked teachers to administer surveys in classrooms on paper. We asked children clarifying questions to assess how well they understood survey questions. Some children misunderstood how to rank species (e.g., they gave all species a 1 or 5) and in subsequent years, we instructed teachers to verbally explain this question when administering surveys. Children sometimes asked questions about what was considered an animal and in subsequent surveys we instructed teachers to tell children to include only non-human, extant animals. We found no difficulties for children in answering any other questions and therefore continued to use the survey data from all years. After the first year, we moved the survey online using Qualtrics, and provided teachers with a script to read before administering the survey. We included the year as a random effect in analyses to test for potential differences and removed any responses in which children clearly misinterpreted the question or had incomplete responses.

Children self-reported demographic information including race (Asian, African American, Caucasian, Hispanic or Latino, Native American, and other), gender, and grade. In 2015, we started asking children whether they or anyone in their family hunted as hunting can influence children' exposure to, and knowledge of local biodiversity (*Peterson et al., 2017*). We also collected school-level socioeconomic data by calculating the percentage of children eligible for free and reduced lunches from the National Center for Education Statistics (https://nces.ed.gov). We distributed permission slips with

information about the study to parents/guardians for schools that required written consent. For schools that did not require written consent, we distributed informational sheets for parents/guardians to opt their children out of the study. Survey methods were reviewed and approved by the North Carolina State University Institutional Review 159 Board for the Protection of Human Subjects (application #4166).

## Data analysis

We coded animals children listed first to their taxonomic class, with the following modifications: fish species were all grouped into one class (fish) and invertebrates were grouped into marine and terrestrial invertebrates. We placed humans, animals that are not real (mythical animals), extinct animals, and the written response "none" into separate categories. We classified each extant, non-human species as local, domestic or exotic (species were assigned one category only). Domestic species included livestock: cows, horses or ponies, sheep, goats, swine, and poultry (chicken and turkey, https://www.nal.usda.gov/nal/animals-and-livestock). Domestic pets included dogs and cats, and the categories of specialty and exotic animals listed by the American Veterinary Medicine Foundation (fish, ferrets, rabbits, hamsters, guinea pigs, gerbils, turtles, snakes, lizards). We used the same criteria described above to identify species as local (i.e., children might have an opportunity to see locally). Some responses children listed were generic to geographic location (e.g., bears, birds) and could have referred to both local and exotic species. We classified these species as local as they fit the definition for local, and children could have the opportunity to view such species, but may not know the specific species name. Any species that did not meet the definition for local or domestic was considered exotic.

Schools were considered suburban, exurban, or rural based on the Silvis housing density categories: suburban (147.048–1,000 houses/km$^2$), exurban (12.64–147.047 houses/km$^2$), rural (0.51–12.63 houses/km$^2$) (Hammer et al., 2004). All analyses were conducted in the RStudio (RStudio, 2015). We tested for significant differences across the variances and means of local animals for liked, scary, and ranked responses using the Fligner-Killeen and the Kruskal–Wallis tests respectively. For significant results, we used a Tukey and Kramer (Nemenyi) test to determine which treatments were significantly different from each other. For the question on how children ranked species, we also conducted a Spearman rank correlation test between the overall rank of species according to each level of urbanization and applied a Bonferroni correction for multiple tests. Before running models, we tested for correlations between categorical covariates using Goodman and Kruskal's tau with the package GoodmanKruskal, and removed the covariate whether a student's family hunts as this was correlated with hunting (>0.50).

We ran three generalized linear mixed models with family set as Poisson using package lme4. The response variables for the three models included the (1) number of local animals children free listed as those they liked, (2) the number of local animals children free listed as scary, and (3) the number of local animals included in children's top five when asked to rank animals (0–5, Table 1). Before running models, we removed responses where the children did not specify a gender as there were very few surveys with no response to this question ($n = 23$), and this was largely due to children running out of time

**Table 1 Summary table of the covariates included in the three models, their estimates, and *p*-values for animals students liked, thought were scary, and ranked from a list.**

| Variable | Liked animals | | Scary animals | | Ranked animals | |
|---|---|---|---|---|---|---|
| | Estimate | *p* | Estimate | *p* | Estimate | *p* |
| Gender | | | | | | |
|   Male | ref | ref | ref | ref | ref | ref |
|   Female | −0.400*** | 0.000 | 0.116** | 0.002 | −0.044 | 0.194 |
| Grade | | | | | | |
|   Grade 4 | −0.946 | 0.108 | −0.621* | 0.016 | 0.018 | 0.928 |
|   Grade 6 | 0.230** | 0.004 | −0.043 | 0.347 | 0.113** | 0.008 |
|   Grade 7 | 0.065 | 0.505 | −0.078 | 0.142 | 0.050 | 0.312 |
|   Grade 8 | ref | ref | ref | ref | ref | ref |
| Race | | | | | | |
|   White | 0.154* | 0.025 | 0.086* | 0.025 | 0.062 | 0.081 |
|   Non-white | ref | ref | ref | ref | ref | ref |
| Hunting | | | | | | |
|   Hunter | 0.330*** | 0.001 | 0.081 | 0.166 | 0.182*** | 0.001 |
|   Not a hunter | ref | ref | ref | ref | ref | ref |
|   No response on hunting | 0.044 | 0.560 | 0.007 | 0.865 | 0.001 | 0.967 |
| Housing Density | 0.000 | 0.276 | 0.000 | 0.082 | 0.000 | 0.255 |
| Proportion of free and reduced lunches | −0.044 | 0.806 | −0.062 | 0.541 | 0.104 | 0.253 |

**Notes:**
* Indicates $p \leq 0.05$.
** $p \leq 0.005$.
*** $p \leq 0.0005$.
na refers to covariates not included in the top final models, and ref are covariates used as a reference level.

(other responses were incomplete). In initial models, race was not a significant factor. Due to the small sample size of some races, we collapsed all races into white (children who only checked Caucasian) and non-white categories (children who checked at least one non-white race category) for final models. Random effects included the school the student attended and year the survey was taken, while fixed effects included the following: gender, race, housing development, the percentage of free or reduced lunches, and hunting (Table 1). For all models, random effects estimates were <0.001 and we therefore proceeded with final models run as generalized linear models in the package MuMIn in R. We ran all combinations of all covariates and considered top models to be those within two AIC points (*Burnham & Anderson, 2002*). We determined coefficient values and significant covariates from the top model or model averaged if there was more than one.

## RESULTS

We implemented surveys in one private and 21 public schools in North Carolina. We included 15 suburban, six exurban, and one rural school. Some schools were sampled multiple years, but with different children. In total, across these schools we collected data from 2,759 children (Table 2). Teacher participation was distributed across schools located in areas ranging from 8.52 to 482.43 houses/km². We sampled fewer rural and exurban schools, in line with the demographics of the state in which more students and schools

**Table 2 Sample sizes of students' responses by covariate categories.**

| | Liked animals | | | Scary animals | | | Ranked animals | | |
|---|---|---|---|---|---|---|---|---|---|
| | Suburban | Exurban | Rural | Suburban | Exurban | Rural | Suburban | Exurban | Rural |
| Total | 1,209 | 862 | 105 | 1,183 | 854 | 107 | 1,032 | 707 | 93 |
| Male | 584 | 397 | 52 | 571 | 390 | 56 | 495 | 325 | 49 |
| Female | 616 | 452 | 52 | 603 | 452 | 50 | 537 | 382 | 44 |
| Grade 4 | 20 | 0 | 0 | 20 | 0 | 0 | 15 | 0 | 0 |
| Grade 6 | 239 | 344 | 0 | 236 | 356 | 0 | 193 | 259 | 0 |
| Grade 7 | 290 | 0 | 105 | 289 | 0 | 107 | 252 | 0 | 93 |
| Grade 8 | 660 | 517 | 0 | 638 | 498 | 0 | 572 | 447 | 0 |
| White | 532 | 360 | 62 | 526 | 361 | 61 | 479 | 326 | 60 |
| Non-white | 677 | 502 | 43 | 649 | 482 | 46 | 553 | 381 | 33 |
| Hunter | 93 | 123 | 58 | 91 | 126 | 58 | 78 | 100 | 52 |
| Not a hunter | 696 | 295 | 47 | 692 | 303 | 49 | 595 | 240 | 41 |
| No response on hunting | 420 | 444 | 0 | 400 | 425 | 0 | 359 | 367 | 0 |

Note:
SES is socioeconomic status. Sample sizes vary across questions because not all students answered all questions or answered questions incorrectly (as in the ranking question).

are in urbanized areas. Our dataset included children from all races and the entire range of socioeconomic status (0–100% of children qualify for reduced/free lunches).

Children listed 8,630 and 8,280 responses (up to four responses per student) for animals they liked and thought were scary, respectively. After removing humans, "none", animals that were not real, extinct animals, and responses we could not decipher, 8,477 responses of liked animals and 8,049 of scary animals were useable for analyses. Of these freeform responses, 24.9% consisted of local animals, 43.3% exotic, and 31.5% domestic. For the ranking question, most children (67%) ranked species according to the directions and incomplete responses or incorrect ranks were removed.

## Freeform responses

Collectively, in the freeform responses where children could write any animal they liked or thought was scary, most children wrote mammals. The most frequently mentioned mammals, regardless of whether they were liked or thought of as scary, were dogs (16.0%), cats (8.2%), pandas (5.6%), rabbits (4.4%), and wolves (3.8%, Fig. 1). For animals that children liked, mammals were recorded nearly twice as often as other animal classes (82.5%, Fig. 2). Birds were the second most frequently mentioned taxonomic class liked (5.8% of listed animals), followed by reptiles (5.2%), fish (3.8%), and terrestrial invertebrates (1.3%; Fig. 2).

Of the animals listed as liked, 44.4% were exotic, 43.5% were domestic, and only 12.1% were local. Suburban children listed the most exotic animals making up 46.04% of their responses, and included a large percentage of domestic animals (41.7%), but few local (12.22%) animals. Similarly, the responses of exurban and rural children consisted mostly of domestic (44.8% and 53.48%) and exotic animals (43.8% and 31.4%) with few local (11.47% and 15.11%, Fig. 3) animals. We allowed children to list up to four species

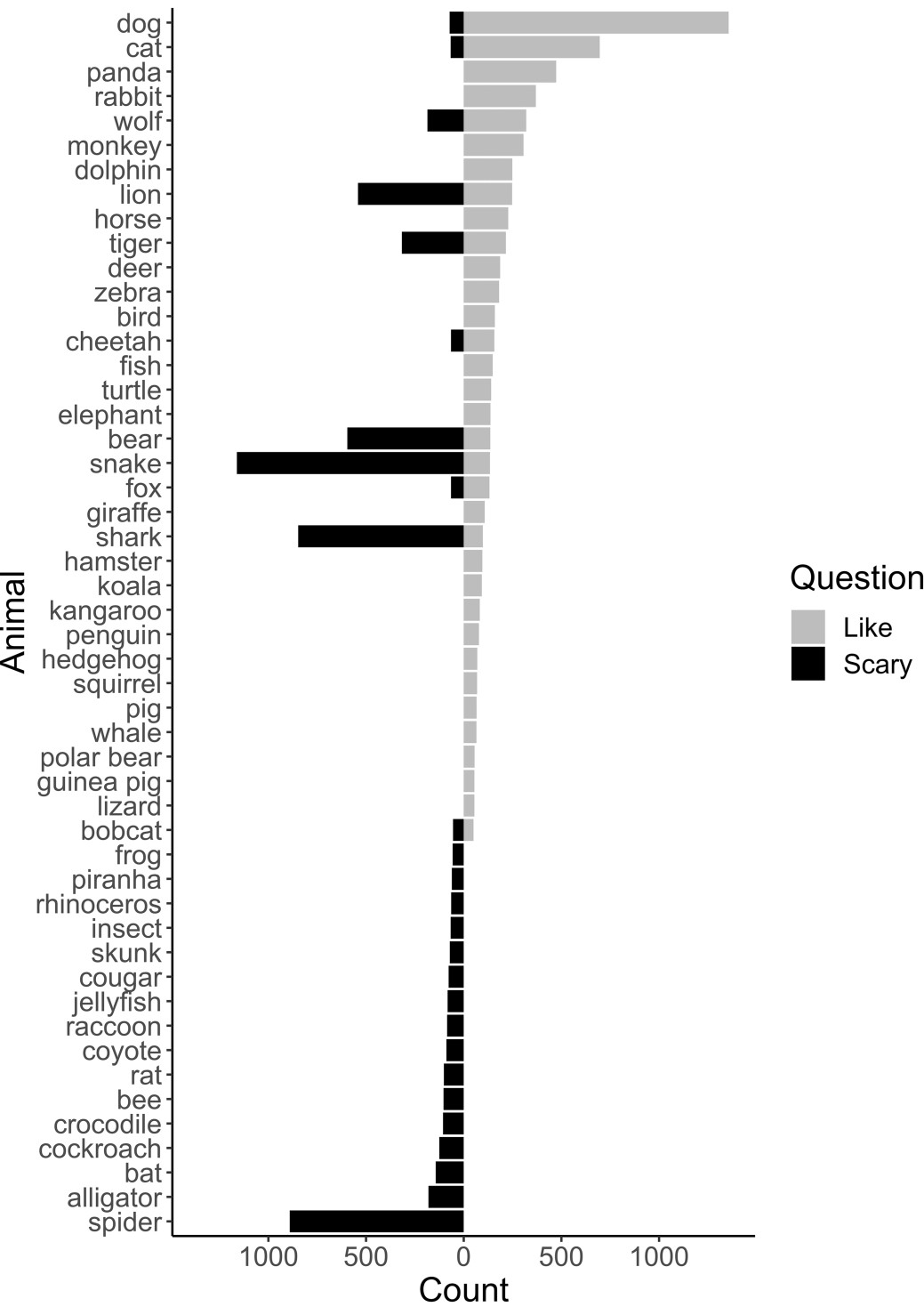

**Figure 1 Free list responses of children listing animals they liked or thought were scary.** Light gray bars show the number of responses for "liked" animals and black represent listing as "scary".

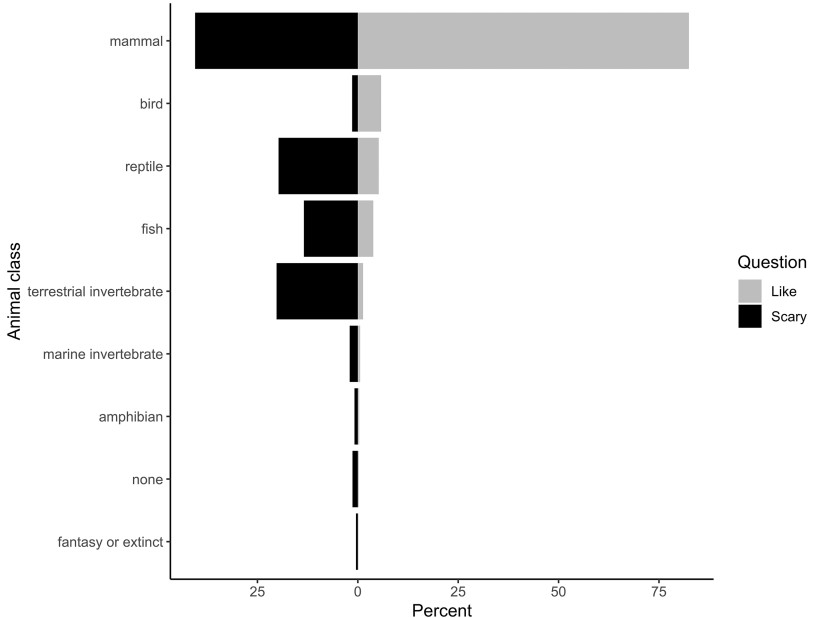

**Figure 2 Classification of animals (modifications noted in text) free listed by children as "liked" or "scary".** Only responses with 50 or more counts are included. Light gray bars show the proportion of responses for "liked" animals and black bars represent listing as "scary".

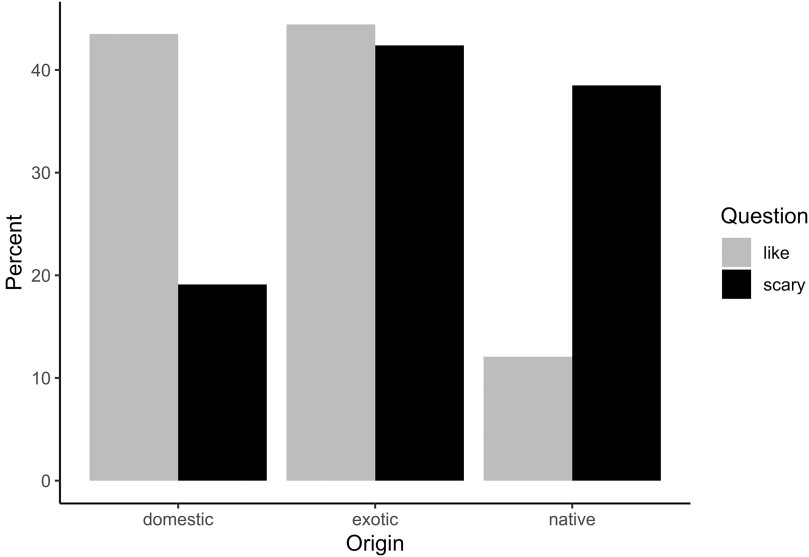

**Figure 3 Percent of local, exotic, and domestic animal animals children free listed as liked or scary.** Light gray bars show the number of responses for "liked" animals and black bars represent listing as "scary".

they liked and on average suburban children included 0.45 (±0.68 SD) local animals, exurban children listed 0.41 (±0.68 SD), and rural children added 0.57 (±0.86 SD) local species (Fig. 4). A Fligner-Killeen test found no significant differences among these groups of children in the variance in the proportion of the animals that they listed that were

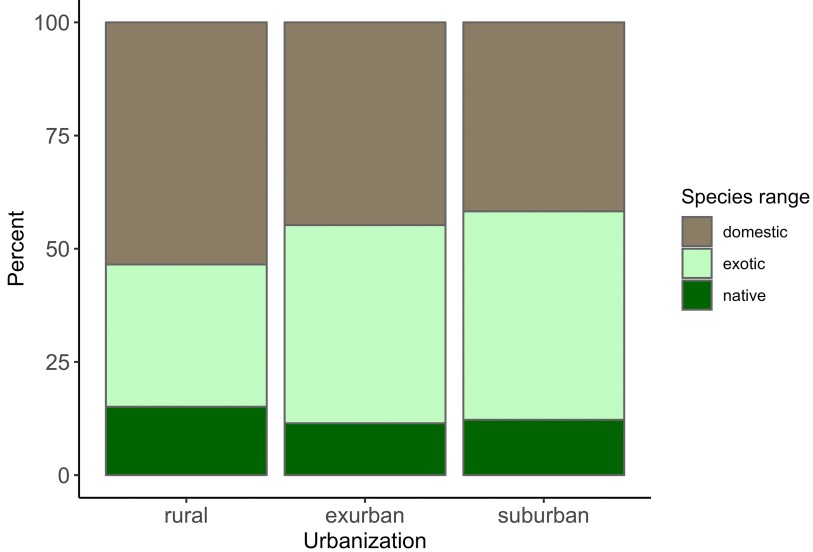

**Figure 4 Percent of "liked" animals that were local, exotic, and domestic grouped by the school location in either a suburban, exurban, and rural areas.**

local ($\chi^2$ = 5.61, d$f$ = 2, $p$ = 0.06) and a Kruskal–Wallis test found no significant differences across the means ($\chi^2$ = 4.88, d$f$ = 2, $p$ = 0.09).

Mammals were also the dominant class for animals considered scary, but whereas nearly all liked taxa were mammals, fewer than half of scary taxa were mammals (40.5%, Fig. 2). Children listed terrestrial invertebrates as the second most scary class of animals (20.23%), followed by reptiles (19.7%), fish (13.4%), and marine invertebrates (2.02%). Children listed almost the same percentage of exotic animals as scary (42.4%) that they listed as liked (44.4%). However, scary animals included far fewer domestic animals (19.1%), and a higher percentage of local animals (38.5%) than the liked species listed for animals children liked. In other words, local, non-domesticated animals were more than three times as likely to be mentioned by children as scary than as liked. When children were asked to free-list four scary animals, on average, suburban children included 1.47 (±1.03) local animals, exurban children listed 1.42 (±0.99), and rural children listed 1.32 (±0.91). A Fligner-Killeen test found no significant differences across variances ($\chi^2$ = 0.04, d$f$ = 2, $p$ = 0.11) and a Kruskal–Wallis test found no significant differences across the means ($\chi^2$ = 2.61, d$f$ = 2, $p$ = 0.27).

## Animal ranking results

Of the five animals ranked from the provided list of 20, the most favorably ranked were all exotic (except for the rabbit); they included the panda, wolf, monkey, and lion (Fig. 5). Animals least often included in children's top five were almost all local animals including the opossum, skunk, raccoon, and bat. The one exception was the rhinoceros, which was the only exotic animal least often included in children's top five. On average, children included 1.94 (±1.04 SD) local animals in their top five rankings with suburban children listing 1.93 (±1.01 SD) local animals, exurban children listing 1.90 (±1.08 SD), and rural children listing 2.33 (±1.11 SD) local animals. We found significant differences in

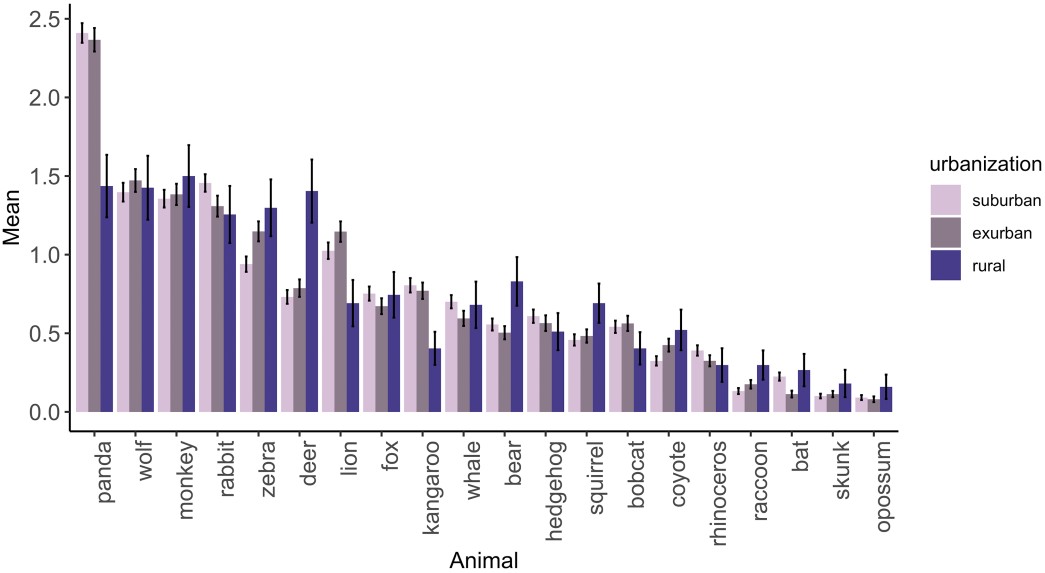

**Figure 5** **The mean ranking of each animal in a provided list of 20 by children across different levels of urbanization.** Higher scores indicate the ranking of animals more favorably. Error bars represent standard deviation of the mean.

variance ($\chi^2$ = 7.85, d$f$ = 2, $p$ = 0.02) using Fligner-Killeen test and log transformed responses after adding one to perform a Kruskal–Wallis one-way ANOVA. These results were also significant ($\chi^2$ = 15.19, d$f$ = 2, $p$ < 0.00) and a post hoc Tukey and Kramer (Nemenyi) test revealed significant differences between suburban and rural ($p$ = 0.00), and rural and exurban areas ($p$ < 0.00) in the average number of local animals ranked. Rural children ranked the panda lower and had higher rankings of most local animals (Fig. 5). However, the Spearman rank correlation test found that the actual rankings of animals, the way the children ordered animals from most favorite to least, was significantly correlated among all levels of urbanization (rural and suburban, $S$ = 222, $p$ < 0.00, $\rho$ = 0.83; rural and exurban, $S$ = 168, $p$ < 0.00, $\rho$ = 0.87; and suburban and exurban, $S$ = 24, $p$ < 0.00, $\rho$ = 0.98), suggesting that children across rural, exurban, and suburban areas rank mammals similarly.

## Model results

For animals that children liked, children who hunted included more local animals ($p$ = 0.001), as did white children ($p$ = 0.025), and children in sixth grade ($p$ = 0.004, Table 1) compared to children that didn't hunt, were non-white, and enrolled in other grades. Female children included fewer local animals ($p$ = 0.000) than males for animals they liked. For scary animals, female and white children recorded more local animals as scary compared to male children and non-white children ($p$ = 0.002 and 0.025 respectively). Only fourth grade children listed fewer local animals as scary than did the other grades ($p$ = 0.016, Table 1). When children were asked to rank animals, children that hunted ($p$ = 0.001) or those who were in grade six ranked more local animals in their top five ($p$ = 0.008, Table 1) than those who didn't hunt or were in other grades.

## DISCUSSION

The similarity of children's categorization of animals across different levels of urbanization suggests that the presumed higher levels of familiarity children in more rural areas have with local wildlife is limited. While we did find that children in rural areas "liked" more local animals, and listed fewer local animals as scary, we also found that youth in exurban, and not suburban areas, "liked" the fewest local animals. However, these differences were marginal, and housing density was not found to be an important factor in the model results when we controlled for student demographics. In short, children across all levels of urbanization viewed wildlife in similar ways. We offer two possible explanations. First, children's exposure to local wildlife species by living in more undeveloped areas may not necessarily translate to more favorable wildlife perceptions or knowledge of local species. Our results may instead suggest that other factors are important in shaping how children perceive wildlife, for instance, outdoor recreation (*James, Bixler & Vadala, 2010*) and cultural norms (*Pease, 2011*). Another possibility is that despite the higher levels of undeveloped land found in rural vs suburban areas, children may not be interacting with it. This latter explanation is supported by mounting evidence that even the most rural children spend more and more time indoors (*Larson et al., 2018*).

In general, local animals made up a larger percentage of perceived scary animals, while they rarely showed up for animals that students liked. This could reflect low knowledge of native biodiversity, which was also found in a previous North Carolina study, in which children listed their favorite animals in North Carolina and the world (*Peterson et al., 2017*). Of these, 87% of the global species were correctly identified as wildlife (e.g., non-pet), but only 60% were correctly identified as native (*Peterson et al., 2017*). In this study, students included a higher percentage of local animals for those that they thought were scary, which suggested that children were aware of and could recall local species. Children frequently listed snakes, spiders, sharks, and bears as scary, and all of these taxa are found in North Carolina. However, emergency room visits for dog bites, the most "liked" animal in this study, were nearly seven times greater than those for venomous snakes and spiders combined and four times more than other uncategorized animals combined (*Langley, 2012*). This mismatch between actual and perceived risk may be explained by negative portrayals of these types of wildlife in the media (*Muter et al., 2012*; *Peterson et al., 2010*). Given the increase in screen time paired with the decrease in time outdoors (*Larson et al., 2018*), it is plausible that children have infrequent encounters with local wildlife, and experiences are primarily virtual.

The modestly higher numbers of local animals listed by rural children may have been more heavily influenced by hunting rather than living in a rural area. Time spent outdoors hunting can provide exposure to local biodiversity, increasing the number of animals students could list and potentially dampen their fears of wildlife. Indeed, *Peterson et al. (2017)* found hunting to be a positive predictor of biodiversity knowledge among children in North Carolina. In our model, children who hunted or who had family members that hunted were more likely to free list local animals for those that they liked and rank them more favorably. Furthermore, rural children ranked deer, an important game species,

as their most liked species, which may be a result of the larger number of hunters among rural children. A total of 55% of rural children hunted compared to 13.8% and 7.6% for exurban and suburban schools respectively. Future research should measure time children spent outdoors and the types of activities children engage in to better understand the mechanisms driving relationships between children's exposure to diversity and perceptions of wildlife.

Despite the troubling trends observed with respect to children's unfamiliarity with local wildlife, several encouraging results for conservation emerged. Of the top 10 animals children listed as "liked", five included species or taxonomic groups with one or more species listed as vulnerable or higher conservation status on the International Union for Conservation of Nature and Natural Resources Red List (e.g., panda, wolf, monkey, dolphin, lion, and tiger). Charismatic, "flagship" species have become iconic for conservation, and while controversial, they have increased positive attitudes toward species, and raised money for organizations (*Dietz, Dietz & Nagagata, 1994*; *Smith & Sutton, 2008*). We also found that children listed higher percentages of wild animals and fewer domestics in the free-listed questions than what has been observed in previous studies (*Bjerke, Ødegårdstuen & Kaltenborn, 1998*; *Lindemann-Matthies, 2005*), again suggesting the potential influence of viewing exotic animals in zoos or in the media. This shift is encouraging, as domestic cats and dogs contribute to native species declines (*Doherty et al., 2017*; *Loss, Will & Marra, 2013*), and when native and domestic species are at odds, difficult measures such as the euthanasia of domestic species are sometimes necessary. These methods are often unpalatable to the public (*Peterson et al., 2017*; *Tannent et al., 2010*), and preferences shifting toward wild animals may allow for greater understanding on such controversial management policies.

Model results revealed that efforts to connect children to nature should target girls and non-white children as well as continuing to engage children as they grow older. That these groups seemed to have particularly low familiarity with or view local animals as scary suggests that they are candidates for efforts to ensure broad support for biodiversity conservation in a local context. Similar trends have been found in other studies and calls to engage girls and minorities have existed for decades (*Foster et al., 2013*; *Lopez, Brown & Unger, 2011*; *Stevenson, Peterson & Dunn, 2017*). Our results that younger students included more local animals as liked (sixth graders) and fewer as scary (fourth graders) than other grade levels could be related to curriculum (e.g., both sixth and fourth grades have wildlife-related standards: NC Department of Public Instruction, http://www.ncreportcards.org/src/). However, other studies find that connections to nature and interest in the environment and wildlife decline as children age (*Frew, Peterson & Stevenson, 2016*; *Stevenson et al., 2013*), suggesting that efforts should continue to find ways to engage with older children. Finally, Caucasian children listed more local species for both liked and scary animals, suggesting a higher level of familiarity with local wildlife. This also reflects previous research, which finds that white children generally have higher environmental literacy levels than minority children (*Stevenson et al., 2013*). This has been linked to cultural views of the outdoors and the environment (*Finney, 2006*; *Johnson, Bowker & Cordell, 2004*) and recreation patterns (*Floyd, Taylor & Whitt-Glover, 2009*;

*Shores, Scott & Floyd, 2007*). As suggested by many (*Lopez, Brown & Unger, 2011*; *Stevenson, Peterson & Dunn, 2017*), our results support the need for culturally sensitive opportunities to engage diverse constituents, including children, with local wildlife.

Although we offer these results as a contribution to conversations around the effects of children's diminished exposure to nature, future research should continue to explore these questions with larger and more diverse samples. In our study, rural children were the least represented, and came from one rural school. While our model results found no school effect, future studies with larger sample sizes and more schools are needed to confirm the patterns observed in this study.

## CONCLUSIONS

Our results imply that it may not be urbanization alone that is driving the Extinction of Experience, as the disconnect with wildlife among children spans across areas of urbanization. As conservation biologists, we are encouraged by the large percentage of globally endangered animals included for animals children liked, but find the low knowledge and unfavorable attitudes toward local species troubling.

Previous research, as well as our own results around hunting, suggests that education and recreation can help. *Lindemann-Matthies (2005)* found educational activities that involved children just noticing native plants and animals on the way to school increased their appreciation of and concern for local species' well-being. Species-targeted programs have even increased children's attitudes toward "unlikeable" species (*Ballouard et al., 2012*; *Tomazic, 2011*). A particularly impactful way of increasing exposure to native wildlife may be through nature-based citizen science programs, where active participation in research encourages observations about the environment, increasing participants' knowledge on local biodiversity. Future studies should focus on understanding the role of such intentional activities in connecting children to nature, and design and evaluate culturally responsive ways of doing so. As the disconnect between children and wildlife is perhaps even more pronounced than previously understood, intentionally providing children experiences in nature may be one of the most important actions conservation biologists can take to promote biodiversity conservation among and for future generations.

## ACKNOWLEDGEMENTS

We would like to thank the teachers and school districts for allowing us to implement surveys in their schools. We also thank Mariah Patton, Rebecca Spears, Spencer Stone, Colleen Lippert, Caitlyn Mothes, and Kristen Lewey for help in survey data organization.

### Funding

This work was conducted with funding from the National Science Foundation grant #1319293. The funders had no role in study design, data collection and analysis, decision to publish, or preparation of the manuscript.

## Grant Disclosure

The following grant information was disclosed by the authors:
National Science Foundation: #1319293.

## Competing Interests

The authors declare that they have no competing interests.

## Author Contributions

- Stephanie G. Schuttler conceived and designed the experiments, performed the experiments, analyzed the data, contributed reagents/materials/analysis tools, prepared figures and/or tables, authored or reviewed drafts of the paper, approved the final draft.
- Kathryn Stevenson conceived and designed the experiments, authored or reviewed drafts of the paper, approved the final draft, design survey and questions.
- Roland Kays conceived and designed the experiments, authored or reviewed drafts of the paper, approved the final draft.
- Robert R. Dunn conceived and designed the experiments, authored or reviewed drafts of the paper, approved the final draft, design survey and questions.

## Human Ethics

The following information was supplied relating to ethical approvals (i.e., approving body and any reference numbers):

Survey methods were reviewed and approved by the North Carolina State University Institutional Review 159 Board for the Protection of Human Subjects (application #4166).

## Data Availability

Raw data is available in the Supplemental Files.

## Supplemental Information

Supplemental information for this article can be found online at http://dx.doi.org/10.7717/peerj.7328#supplemental-information.

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
