# Peer review of "Children’s attitudes towards animals are similar across suburban, exurban, and rural areas"

_PeerJ, doi:10.7717/peerj.7328_

## Round 0.1 · original submission · Minor Revisions

This paper provides an interesting and valuable survey of youth attitudes toward nature. I recommend that you revise the manuscript in light of all the reviewer's comments and especially the following:

1. the writing could be improved by revising run-on sentences, such as at line 25, line 383, line 406,
2. consider other terminology rather than "development zones" that might be clearer to biologist readers
3. I am concerned by the lack of "rural" schools in the study. Were there any "urban" or "inner-city" schools? The analysis beginning on line 280 may change if you had more rural students in the study. Perhaps instead of binning the schools based on the housing density, you could just analyze the local housing density as a continuous predictor variable. Please either adjust your analysis to explore this alternative, or explain in your response why you chose to bin the schools.
4. Clean up the writing by revising incorrect grammar and using the past tense throughout. Examples include: "uphold" -> "are upheld" in line 365.
5. The discussion section is too long and it seems to wander. I suggest shortening it to ~ 5 paragraphs, each with a clear topic sentence that highlights the main takeaway points of the paper.

·

Basic reporting

I found no major problem. The general framework is clearly exposed. Method, data and analyses are well presented. Results are convincing. Conclusions are sound.

Experimental design

There is no real experimental design. This study is empirical. Yet large data set provides robustness.

Validity of the findings

The results were expected, the patterns are clear. I fully trust the findings. The statistics were simple, but I do not see any reason to perform sophisticated ones, this will add nothing.

Additional comments

This study reports interesting results based on questionnaire responses obtained from a large sample of children. The method used and the type of questions addressed have been somehow previously employed; but considering that few investigations of this kind have been performed, it is essential to challenge published studies. Indeed, as nicely exposed in the introduction, exploring the knowledge and possible attitudes of schoolchildren toward wildlife is a key pre-requisite to guide educational actions. Unfortunately, the results (expectedly) show that children have a minimal knowledge regarding local fauna, and thus are disconnected from wildlife. This study therefore militates for the necessity to bring children in the field. Although this conclusion might seem obvious or over simplistic, robust study like the current one are very important because the reality is that children spend an increasing amount of time in front of their screen and thus proportionally less in the field. For convenience and for multiple other reasons, many schools encourage this trend. Overall, I enjoyed reading the manuscript. Below I propose several minor comments.

The discussion is too long, somehow repetitive and often commenting detailed results (e.g. line 440-447). Perhaps that the discussion could be more streamlined? Similarly the long conclusion is a summary of the discussion (e.g. kids love dogs). Perhaps providing few recommendations (e.g. schoolchildren should spend more time in the field) or future directions (do we really need umbrella species, like loveable mammals, during outdoor activities?) might be more appropriate.

Line 140: Although I understand why whales or wolfs were categorized as exotic animals (Humpback whales can be observed from North Carolina beaches but the schools were inland), it is less clear for bats (several widespread bat species occur inland in North Carolina). Further, bats were moved later displaced into the local group (line 149), this is confusing.

Line 275: rabbit is not an exotic species (see line 139)

Line 433 etc.: why more efforts should be devoted to the groups displaying “lower” scores? Is the objective to eliminate differences among groups? In practice, it might be complicated and resource consuming because specific actions will be needed. Improving the attitude and knowledge of all kids from a classroom seems to be more easy and likely more beneficial, despite group differences.

Meuser et al. 2009 (line 71) is not in the reference list.

Several typos and formatting problems can be found in the reference list, this should be checked.

Reviewer 2 ·

Basic reporting

The manuscript was clear and easy to read, with few grammatical mistakes and misspellings. I recommend more recent citations, especially in the introduction, to support the authors' claims. For instance, lines 50, 51, and 234. The hypotheses were clearly stated. One suggestion, however, is to use a better term for "development zones." In the context of studying children, this term can be easily confused with biological development. Perhaps economic development would be a less confusing term?

Experimental design

The experimental design seemed realistically simple given the type of data. The research questions were well defined and all were tested and results reported. I was confused by line 124-what was a "control" teachers and where is this discussed in the results? Unfortunately, the study had only a single rural school, which greatly limits the impact of the study and the ability to measure a gradient of the effect of economic development. I did not see where the Filgner-Killeen test was discussed in the methods.

Validity of the findings

I believe that the results and conclusions accurately reflect the findings. There were "negative findings," but that makes the study interesting. I think the authors should discuss the lack of rural schools more, and how that affects the results, conclusions, and future studies.

Additional comments

Overall, I enjoyed this manuscript, but am disappointed that the sample size of rural schools was so low, as it makes for a much weaker impact. In the introduction, the "pigeon paradox" could be explained in more detail, and in the conclusion, it would be nice to see some recommendations on how to combat disconnect with nature.

---

## Round 0.2 · Minor Revisions

I enjoyed reading this revised version. I have noted a few minor edits to the text and figures that will improve the clarity of the paper. Please see the attached, marked up manuscript, for comments.

---

## Round 0.3 · accepted · Accept

Thank you for your revision. The manuscript is a valuable addition to the growing literature on attitudes toward nature in the United States.